# Detection and Classification of Cotton Foreign Fibers Based on Polarization Imaging and Improved YOLOv5

**DOI:** 10.3390/s23094415

**Published:** 2023-04-30

**Authors:** Rui Wang, Zhi-Feng Zhang, Ben Yang, Hai-Qi Xi, Yu-Sheng Zhai, Rui-Liang Zhang, Li-Jie Geng, Zhi-Yong Chen, Kun Yang

**Affiliations:** 1School of Physics and Electronic Engineering, Zhengzhou University of Light Industry, Zhengzhou 450002, China; 2Fiber Inspection Bureau in Henan Province, Zhengzhou 450002, China

**Keywords:** deep learning, foreign fiber detection, YOLOv5, polarization imaging, line laser

## Abstract

It is important to detect and classify foreign fibers in cotton, especially white and transparent foreign fibers, to produce subsequent yarn and textile quality. There are some problems in the actual cotton foreign fiber removing process, such as some foreign fibers missing inspection, low recognition accuracy of small foreign fibers, and low detection speed. A polarization imaging device of cotton foreign fiber was constructed based on the difference in optical properties and polarization characteristics between cotton fibers. An object detection and classification algorithm based on an improved YOLOv5 was proposed to achieve small foreign fiber recognition and classification. The methods were as follows: (1) The lightweight network Shufflenetv2 with the Hard-Swish activation function was used as the backbone feature extraction network to improve the detection speed and reduce the model volume. (2) The PANet network connection of YOLOv5 was modified to obtain a fine-grained feature map to improve the detection accuracy for small targets. (3) A CA attention module was added to the YOLOv5 network to increase the weight of the useful features while suppressing the weight of invalid features to improve the detection accuracy of foreign fiber targets. Moreover, we conducted ablation experiments on the improved strategy. The model volume, mAP@0.5, mAP@0.5:0.95, and FPS of the improved YOLOv5 were up to 0.75 MB, 96.9%, 59.9%, and 385 f/s, respectively, compared to YOLOv5, and the improved YOLOv5 increased by 1.03%, 7.13%, and 126.47%, respectively, which proves that the method can be applied to the vision system of an actual production line for cotton foreign fiber detection.

## 1. Introduction

Cotton is the largest natural fiber in the textile industry. During the processes of cotton cultivation, harvesting, transportation, and processing, a large number of foreign fibers is inevitably mixed in due to various factors, such as cotton hulls, leaves, mulch films, chemical fibers, and paper pieces. These foreign fibers have adverse effects on the textile products, not only reducing the spinning efficiency, but also causing fabric defects and reducing product grade [1]. Therefore, the detection of foreign cotton fibers is an important and necessary step before spinning. It is time-consuming and inefficient to rely on workers to manually detect foreign fibers from cotton, and the detection accuracy of foreign fibers is low [2,3]. In recent years, numerous detection methods for foreign fibers have been developed, including photoelectric, ultrasonic, and optical detection, according to the detection principle [4,5]. However, photoelectric detection technology can only detect colored foreign fibers but not white transparent foreign fibers [6]. Ultrasonic detection technology can only detect foreign fibers in a large area, and its speed is slower [7]. Presently, foreign fiber detection mainly uses machine vision technology with high recognition rate, high detection speed, and low cost. Machine vision technology can be divided into X-ray imaging technology, ultraviolet fluorescence imaging technology, infrared imaging technology, line laser imaging technology, hyperspectral imaging technology, and polarized light imaging technology [8]. Pai et al. [9] identified and classified three types of foreign fibers in cotton using an X-ray microtomographic imaging system. However, the imaging speed of the X-ray imaging method is slow, and the equipment cost is high. Luo et al. [10] proposed a machine vision method combined with UV fluorescence to sort foreign fibers in cotton. However, UV fluorescence imaging is less effective in detecting foreign fibers that are similar to cotton in color and without a fluorescence effect. Cai et al. [11] imaged cotton using 12 types of foreign fibers in the near-infrared band. The infrared imaging method has a better detection effect for foreign fibers with a significant difference in the absorption between the near-infrared band and cotton; however, the infrared spectrum camera is slow and expensive, and the relevant technology is still in the laboratory research stage. Hua et al. [12] proposed a method to identify foreign fibers based on line laser imaging; solid-state lasers have been widely used in machine vision detection owing to their low cost, small volume, and ease of operation. Mustafic et al. [13] employed hyperspectral fluorescence imaging to identify foreign fiber in cotton; however, hyperspectral imaging technology requires a high external environment, and the devices are expensive. Zhang et al. [14] utilized polarization imaging technology to increase the detection rate of transparent films.

The foreign fiber detection algorithm is the core part of foreign fiber recognition and classification and can be divided into traditional image algorithms and deep learning image algorithms. Traditional image algorithms rely on the artificial design of foreign fiber characteristics by the algorithm designer and utilize image preprocessing, feature extraction, feature selection [15,16,17], image segmentation [18,19], and image classification [20,21,22] to achieve foreign fiber detection. However, traditional image algorithms have limited ability to recognize and classify multiple types of foreign fiber targets and cope with complex scenes, whereas deep learning image algorithms have the ability to learn excellent complex features. He et al. [23,24,25] achieved the recognition of foreign fibers in seed cotton images based on a Faster-RCNN. Du et al. [26] and Dong et al. [27] used ResNet-50 and Inception-ResNet-V2 instead of the original VGG16 of Faster-RCNN to extract the features of foreign fibers, and the K-means++ algorithm was used to improve the size and number of candidate boxes to achieve the classification and localization of multiscale foreign fibers. Wu et al. [28] introduced the MobileNets network and constructed the MobileNets YOLOv3 model to detect foreign fibers in cotton. Wei [29] implemented a real-time intelligent classifier for foreign fiber images. On a dataset of 20,000 foreign fiber images, the classification accuracy reached 95%. Wu et al. [30] combined traditional convolution with depth-wise separable convolution and introduced a convolutional layer attention mechanism to establish a deep learning model for recognizing foreign fibers in cotton. The recognition accuracy for five types of foreign fibers, such as plastic ropes and human hairs, was 91.93% on the test set. Zhang et al. [31] introduced the residual network as a feature extraction network and combined it with the feature pyramid to propose an improved Faster R-CNN network for the detection of six types of foreign fibers, such as feathers and waste paper. The accuracy and recall rate of this network were 97.6% and 82.4%, respectively, which were higher than those of the VGG16 and ResNet50 networks. Zhang et al. [32] utilized the YOLOv5 neural network to perform classification and recognition of weeds, blackjack, and other foreign fibers that were segmented from images. The content of various foreign fibers was also measured, and the recognition accuracy reached 98%.

The actual production line of cotton requires an extremely high detection speed and a lightweight network with a smaller volume and faster detection speed. Moreover, cotton on the actual production line is carded through a carding machine, and foreign fibers mixed in the cotton are broken into smaller foreign fibers. These methods fail to consider the effective detection of foreign fibers in small targets. Therefore, this study is based on the YOLOv5 algorithm, and improved methods of Shufflenetv2 and PANet are introduced into YOLOv5. An improved YOLOv5 algorithm combined with an attention mechanism module (CA) network is proposed in this paper, which can realize the real-time detection of foreign fibers of multiple types of small targets. 

The following contributions are made by our work:A polarization imaging device of cotton foreign fiber was constructed using line laser polarization imaging technology.In order to reduce the model volume and improve the detection speed, the lightweight network Shufflenetv2 with Hard-Swish function was added as the backbone feature extraction network.In order to increase the detection accuracy of foreign fibers in small targets, an improved PANet was added to YOLOV5.The CA module was added before the Head of YOLOv5 to allocate the weight of the channel features and spatial features to improve the accuracy of foreign fiber recognition and classification.

In summary, the line laser polarization imaging approach proposed in this study has an important guiding value for the online identification and classification of cotton foreign fibers and the control of foreign fiber generation in cotton planting and picking. Compared with other typical object detection algorithms, our proposed algorithm has a higher detection speed, smaller model size, and higher detection accuracy and is more suitable for foreign fiber detection tasks. 

## 2. Materials and Methods

### 2.1. Experiment Materials

The cotton and foreign fiber samples used in the experiment were provided by the Henan Fiber Inspection Bureau and originated from the Xinjiang Uygur Autonomous Region, China. The experiment was conducted using 20 common types of foreign fibers in cotton, as shown in Figure 1, and the sizes of foreign fibers were categorized as 0.5 mm^2^, 1 mm^2^, 1.5 mm^2^, 3 mm^2^, and 5 mm^2^. Group 1 comprised colored foreign fibers, and it was easier to distinguish them in cotton, whereas Group 2 comprised white transparent foreign fibers that were more difficult to detect because they are extremely similar to cotton fiber in color and appearance.

### 2.2. Experiment Equipment

In actual detection, cotton containing foreign fibers was first made into a thin layer with a width of approximately 10 cm and thickness of approximately 2 mm. The cotton thin layer sample was irradiated by a uniform line laser, and the scattered light of cotton was mist-like. Mulch film, plastic and paper pieces, and other white foreign fibers are mostly dense materials, and the reflected light is approximately a mirror reflection [12]. 

The experiment found that the characteristic information of cotton foreign fiber image was the most prominent when the incident angle of the line laser was about 45°. For example, when the laser incident angle was 45°, the average gray value (M(X)) of the foreign fiber image was larger, and the contrast value (Var(x, y)) was the largest, as shown in Table 1.

Because of the different polarization characteristics of different foreign fibers, the reflected light waves have polarization information of the foreign fibers, and different types of foreign fibers can be distinguished through polarization imaging [14].

A physical image of the cotton foreign fiber polarization imaging detection device is shown in Figure 2. The sensor of the camera (MV-CH050-10UP, HIKROBOT) was equipped with four-way (0, 45, 90, 135) pixel-level polarization filters with a resolution of 2448 × 2048 and a target surface size of 2/3″ using USB power output. The light source was a 405 nm line laser (SL-405-35-S-B-90-24V, OSELA) with a power of 35 mW. 

### 2.3. Dataset, Environment, and Parameters

The target detection dataset in this study was acquired using the image acquisition system shown in Figure 2, containing a total of 3944 foreign fiber target images of 20 categories, which were divided into training, validation, and test sets. The data were enhanced by gaussian blur, affine transformation, brightness transformation, dropping pixel transformation, and flip transformation [33,34]. The enhanced dataset consisted of 21,381 images, and the data format was JPG. Table 2 lists the statistical information of the dataset.

The hardware environment and software versions of the experiments are listed in Table 3.

In this study, the SGD (stochastic gradient descent) method was used to optimize the learning rate, and the epochs were determined by comparing the loss functions of the training set and validation set. The parameters of the training network are listed in Table 4. 

### 2.4. Loss Function and Model Evaluation Metrics

The loss function of YOLOv5 consists of three components, which are confidence loss, bounding box regression loss, and classification loss. The expression of the YOLOv5 loss function is shown below:(1)$$Loss_{total}=\lambda_{1}L_{obj}+\lambda_{2}L_{box}+\lambda_{3}L_{cls}$$

*L*_*obj*_, *L_box_*, and *L_cls_* represent confidence loss, bounding box regression loss, and classification loss, respectively. *λ*_1_, *λ*_2_, and *λ*_3_ are weight coefficients for the three losses, and changing these coefficients can adjust the emphasis on the three losses. In YOLOv5, *L_box_* is calculated using *L_CIoU_* [35], which can improve both the speed and accuracy of bounding box regression. The expression for *L_CIoU_* is shown below:(2)$$\left\{\begin{array}{c} L_{CIoU}=1-IoU+\frac{\rho^{2}\left(b,b^{gt}\right)}{c^{2}}+\alpha v\\ IoU=\frac{\left|b\cap b^{gt}\right|}{\left|b\cup b^{gt}\right|}\\ v=\frac{4}{\pi^{2}}{\left(\arctan\frac{w^{gt}}{h^{gt}}-\arctan\frac{w}{h}\right)}^{2} \end{array}\right.$$

In the above expression, *b* and *b*^*gt*^ represent the predicted box and ground truth box, respectively; *w*^*gt*^, *h*^*gt*^, *w*, and *h* represent the width and height of the ground truth box and the predicted box, respectively; *ρ* represents the distance between the centers of the two boxes; *c* represents the maximum distance between the boundaries of the two boxes; and *α* is a weight coefficient. Both *L_obj_* and *L_cls_* use the BCEWithLogitsLoss, and their calculation formula is shown below:(3)$$Loss=-\frac{1}{n}\sum \left[y_{n}\ln x_{n}+\left(1-y_{n}\right)\ln\left(1-x_{n}\right)\right]$$

The BCEWithLogitsLoss function includes both the Sigmoid layer and the BCELoss layer and is suitable for multi-label classification tasks; *y_n_* represents the ground truth label, and *x_n_* represents the predicted label.

To verify the superior performance of the improved Yolov5 model, we measured the *mAP*, *FPS*, model volume, etc. Some commonly used metrics of precision (*P*), recall (*R*), average precision (*AP*), *F*1 Score (*F*1), and mean average precision (*mAP*) were selected to evaluate the model performance [36], and the metrics were defined as follows:(4)$$precision=\frac{TP}{TP+FP}$$
(5)$$recall=\frac{TP}{TP+FN}$$
(6)$$F1=\frac{2\cdot precision\cdot recall}{precision+recall}$$
(7)$$AP=\int_{0}^{1}p\left(r\right)dr$$
(8)$$mAP=\frac{\sum_{i=1}^{N}AP_{i}}{N}$$

*TP* denotes the positive samples predicted to be correct, *FP* denotes the negative samples predicted to be incorrect, *FN* denotes the positive samples predicted to be incorrect, and *N* denotes the number of sample categories.

### 2.5. Improvement of YOLOv5 Network Architecture

#### 2.5.1. YOLOv5 Network Architecture

YOLOv5 combines the characteristics of YOLOv1, YOLOv2, YOLOv3, and YOLOv4. YOLOv5 mainly contains four network models, namely, YOLOv5s, YOLOv5m, YOLOv5l, and YOLOv5x, and the model size and parameters increase sequentially in the four network structures. This study was based on the YOLOv5s network structure, as shown in Figure 3.

The YOLOv5 network structure consists of a backbone, neck, and head, and the image input first goes through the backbone for continuous feature extraction. The focus performs a slice operation on the input image; for example, if the input image size is 640 × 640 × 3, the slice operation will take a value for every other pixel on the image, and the result will be stacked on the channel to obtain a feature layer of 320 × 320 × 12. It is commonly understood to expand the image channel and compress the image height and width. The focus-module structure is shown in Figure 4.

The second layer of the backbone is the CBS module with a convolution kernel size of 3 × 3, which performs convolution the calculation, batch standardization calculation, and SiLU activation function on the input data, adds nonlinearity to the network, and accelerates the convergence speed of the network. The third layer is the C3 module, which is mainly composed of n bottleneck modules, three CBS modules, and two convolution layers of size 1 × 1, it and is designed to better extract the deep features of the image. The structures of the bottleneck and C3 modules are shown in Figure 5 and Figure 6, respectively.

The last layer of the backbone is the SPP module. First, the number of channels of the input image is halved using the first CBS module, and then the feature map output from the first CBS module is passed through three maximum pool layers of different sizes (13 × 13, 9 × 9, and 5 × 5), and the residual edges constructed together with the output of the first CBS module are connected in parallel. Finally, the number of channels is halved by the second CBS module to ensure that the height and width of the feature map of different size inputs can be kept consistent after pooling; the structure of the SPP module is shown in Figure 7.

The neck network constructs feature pyramids for enhanced feature extraction to obtain more contextual information. Three feature maps are generated in the backbone network; the three feature layers are 80 × 80, 40 × 40, and 20 × 20 from shallow to deep. After the three effective feature layers are obtained, the FPN feature pyramid structure is constructed first, and the 20 × 20 feature layer is upsampled to obtain the 40 × 40 feature layer and then stacked with the corresponding 40 × 40 feature layers in the backbone network. A feature layer of 80 × 80 was obtained by upsampling twice in the FPN structure, and strong semantic features were transferred. Subsequently, the PAN structure was constructed to convey stronger localization features, and the 80 × 80 feature layer was downsampled to obtain a 40 × 40 feature layer, which was stacked with a 40 × 40 feature layer in the FPN network structure. The PAN network structure was downsampled twice, and the final outputs were 80 × 80, 40 × 40, and 20 × 20 enhanced effective feature layers, respectively. Finally, we used these three enhanced feature layers to input the Yolo Head to obtain the regression prediction and classification prediction results. 

#### 2.5.2. Proposed Approach: YOLOv5-CFD

This study made corresponding improvements to the backbone, neck, and head of YOLOv5. First, Shufflenetv2 was introduced as the backbone feature extraction network under the premise of ensuring detection accuracy. The weight parameter and volume of the network were reduced, and the lightweight design of the model was realized. Moreover, because the foreign fibers were mostly small-sized targets, the FPN + PAN structure was modified to obtain feature maps with more fine-grained information. Finally, the CA attention module was added to the front of the Yolo Head to improve the detection accuracy. The improved YOLOv5 (YOLOv5-CFD) network structure is illustrated in Figure 8.

##### Improvement of Backbone Network

ShufflenetV2 was proposed by Ma et al. [37] and was based on ShufflenetV1 and four efficient network design principles. The ShufflenetV2 model excels in speed and accuracy, making it an ideal lightweight network for deployment in mobile devices. First, ShufflenetV2 divides the input of the feature channel into two branches by the “Channel Split” operation. One branch has the same structure, and the other branch consists of three convolutions with the same input and output channels. The two branches are concatenated after convolution to keep the number of channels constant. Finally, the “Channel Shuffle” operation is used to ensure the information exchange between the two branches. ShufflenetV2 contains a basic unit and a unit for spatial downsampling (2×), as shown in Figure 9.

In this paper, ShufflenetV2 units with stride = 2 and stride = 1 were chosen to construct a new backbone network, and the output of each stage in the new backbone was connected to PANet. Moreover, we replaced the activation function in the ShufflenetV2 unit with the H-swish activation function, as shown in Equation (9):(9)$$H\text{-}swish\left(x\right)=\left\{\begin{array}{ccc} 0 & & x\leq -3\\ x & & x\geq 3\\ x\cdot \left(x+3\right)/6 & & -3<x<3 \end{array}\right.$$

##### Improvement of PANet Network

Among the three effective features of the FPN + PAN structure output, the 20 × 20 and 40 × 40 feature maps were used to detect larger targets, whereas foreign fibers in cotton are mostly small-sized targets. Moreover, the image size of our input network was 2448 × 2048, and the grid pixels corresponding to the 20 × 20 and 40 × 40 feature maps were 128 × 108 and 64 × 54, respectively, when performing the bounding box regression. The k-means clustering statistics showed that nearly 75% of the foreign fiber target pixels were below 60, as shown in Figure 10, with 20 × 20 and 40 × 40 feature maps corresponding to anchors ([116, 90], [156, 198], [373, 326]) and ([30, 61], [62, 45], [59, 119]). The anchors were larger, and many operations were useless when performing the bounding box regression. The 20 × 20 and 40 × 40 feature maps and large target identification frames were discarded, making the bounding box regression more accurate and minimizing the model computational cost.

To solve the problem of an excessive number of small targets, the PANet network connection was improved to obtain a feature map with fine-grained information. A new 160 × 160 feature map was generated by upsampling the output of the backbone network twice and fusing it with the feature map of the corresponding size of the backbone. Because the improved backbone network generated three layers of feature mapping of 320 × 320, 160 × 160, and 80 × 80, the FPN did not require secondary upsampling; hence, the final YOLO detection heads were 160 × 160 and 80 × 80; Figure 11 shows the PANet network improvement schematic diagram of YOLOv5.

##### CA Module Design

Hou et al. [38] proposed a novel attention mechanism for mobile networks called “Coordinate Attention” by embedding location information into channel attention in 2021, as shown in Figure 12.

Coordinate Attention focuses on the image width and height and encodes precise position information. First, the input feature map was divided into the width and height directions for global averaging pooling to obtain the feature maps in the width and height directions. The output of the *c*-th channel with the height and width is expressed as follows:(10)$$z_{c}^{h}\left(h\right)=\frac{1}{W}\sum_{0\leq i<W}x_{c}\left(h,i\right)$$
(11)$$z_{c}^{w}\left(w\right)=\frac{1}{H}\sum_{0\leq j<H}x_{c}\left(j,w\right)$$

The above equation integrates the features from different directions and outputs a pair of feature maps with known directions. The module can capture long distance relationships in one direction while retaining spatial information in the other, helping the network locate targets more accurately. 

Stitching together the feature maps in the width and height directions of the obtained global perceptual field, the channel is compressed to the original C/r using a 1 × 1 convolution. Subsequently, the BatchNorm and *H-swish* activation functions are used for encoding, followed by a 1 × 1 convolution to adjust the channels of the feature map to be equal to the number of channels of the input feature map. The attention weights *g^h^* and *g^w^* of the feature map on the height and width, respectively, are obtained after the sigmoid function, as shown below:(12)$$g^{h}=\sigma \left(F_{h}\left(f^{h}\right)\right)$$
(13)$$g^{w}=\sigma \left(F_{w}\left(f^{w}\right)\right)$$

Finally, a weighted fusion is performed on the original feature map to obtain the final feature map with attention weights in the height and width directions, as shown in the following equation:(14)$$y_{c}\left(i,j\right)=x_{c}\left(i,j\right)\cdot g_{c}^{h}\left(i\right)\cdot g_{c}^{w}\left(j\right)$$

Based on the characteristics of multiple types and small targets with different fibers, this study added a CA module at the front end of each of the two detection heads of the Yolo Head to improve the performance of the network at a low cost, thus improving the overall accuracy of target detection.

The flow chart of the foreign fiber detection method used in this study is shown in Figure 13.

## 3. Results and Discussion

Figure 14 shows the loss reduction curves of the YOLOv5-CFD model for the training and validation sets of foreign fiber images. As can be observed from the loss curve, the loss value dropped to a relatively small value when the number of training rounds was 20, and the network stabilized when the number of training rounds was 120.

The confusion matrix of the YOLOv5-CFD model is shown in Figure 15. It can be observed from the figure that most of the targets of different fiber types were correctly predicted with a low target miss rate, indicating that the model exhibited good performance.

Figure 16 shows the PR curve of YOLOv5-CFD test set and shows the change curve of the accuracy and recall of the detection results of twenty kinds of foreign fiber targets. According to statistics, the overall detection result mAP@0.5 was 96.9%.

### 3.1. Ablation Experiment

The effect of the improved method on the model performance was analyzed by ablation experiments. For comparison purposes, the experiment was divided into five groups. The first group was the original YOLOv5 network. In the second group, the ShufflenetV2 module was introduced into the backbone feature extraction network module of the YOLOv5. The third group modified the PANet network connection method using YOLOv5. In the fourth group, a CA module was added to the front of each of the two detection heads of YOLOv5. The last set of experiments was the result of the model used in this study. The experimental results are listed in Table 5. 

As seen in Table 5, the use of the ShufflenetV2 module in the back-bone feature extraction network reduced mAP@0.5 and mAP@0.5:0.95 by 1.95% and 2.73%, respectively, but the model volume decreased by 5.89 MB, and the detection speed increased by 200 f/s. The introduction of the ShufflenetV2 module played an important role in reducing the model volume and improving the detection speed. The improvement of the PANet network reduced the model volume by 3.3 MB, increased the mAP@0.5 and mAP@0.5:0.95 by 0.27% and 4.63%, respectively, and increased the detection speed by 153 f/s. The introduction of the CA attention module improved the detection accuracy of the model and verified the effectiveness of the improved PANet and CA modules. In summary, the improved strategy based on YOLOv5 proposed in this study is important for facilitating the identification and detection of cotton foreign fibers in an actual production line.

### 3.2. Comparison of Different Models

To verify the superiority of the YOLOv5-CFD model in cotton foreign fiber detection, we compared it with the most advanced foreign fiber detection models, YOLOv5, YOLOv4, SSD, and Faster-RCNN. The relevant parameters of the experiments were also strictly controlled using a uniform image size as the input and a uniform training and test set for experimental testing. 

Comparing the overall test results of Faster-RCNN, SSD, YOLOv4, YOLOv5, and YOLOv5-CFD with mAP@0.5, as shown in Figure 17, it can be seen that YOLOv5-CFD model had better performance.

The pictures used in the comparative experiment in Figure 18 are from the test set of this paper [39]. Each experiment was conducted in the same environment. Figure 18 shows the detection effects of different models in different cases. The images contain complex light environments, small target foreign fibers, and multiple types of foreign fibers, so the problems of multiple types of small target foreign fibers in a complex light environment are fully considered, providing a convenient way to fully demonstrate the robustness and generalization ability of the model.

From the image detection results, it could be observed that for large foreign fibers, most of the five models were recognized, and YOLOv5-CFD had the highest correct classification rate. For small foreign fibers, YOLOv5-CFD had the highest recognition rate and correct classification rate. For the first image, YOLOv5-CFD was identified and classified correctly. In the second image, YOLOv5-CFD had the highest recognition rate with only one missed target, and YOLOv5 and Faster-RCNN had the highest correct classification rate. For the last image, YOLOv5-CFD, YOLOv5, and Faster-RCNN were all identified correctly, and only YOLOv5-CFD and SSD were classified correctly; however, the SSD model had multiple overlapping detection frames in the detection. In summary, the YOLOv5-CFD model outperformed the other four models in terms of the test results.

As shown in Table 6, the model volume, mAP@0.5, mAP@0.5:0.95, and FPS of the YOLOv5-CFD were up to 0.75 MB, 96.9%, 59.9%, and 385 f/s, respectively, which were better than the values of YOLOv5 (13.82 MB, 95.87%, 52.77%, and 170 f/s, respectively), followed by YOLOv4 (244.78 MB, 93.59%, 50.50%, and 88 f/s, respectively), and SSD (100.29 MB, 83.07%, 39.06%, and 128 f/s, respectively). Furthermore, the results of Faster-RCNN (108.91 MB, 75.68%, 33.60%, and 9 f/s, respectively) were worse. The results showed that the overall performance of the proposed YOLOv5-CFD was the best [40].

The main improvement of the YOLOv5-CFD model is the volume size of the model and the detection speed; these enhancements meet the high requirements of the actual production line detection of cotton foreign fibers, and the detection accuracy of YOLOv5-CFD for small target foreign fibers is also the highest. Based on the above analysis, the YOLOv5-CFD object detection algorithm proposed in this study improves the detection speed and accuracy of foreign fiber targets and significantly reduces the model size.

### 3.3. YOLOv5-CFD Test Results

In order to test the robustness and anti-interference of the YOLOv5-CFD model, this paper repeatedly tested the miss-recognition rate, misjudgment rate, precision, recall, and F1 score of the model under different illumination, different incident angles, different cotton foreign fiber samples, different foreign fiber positions, different foreign fiber sizes, and different environments. Combined with the sampling frequency of the camera, the speed of the conveyor belt was set to 4 m/min. The misrecognition rate is the rate of failure to identify the presence of foreign fibers, and the misjudgment rate is the rate of judging the position where there is no foreign fiber as the presence rate. For each test condition, the precision and recall values for each category are first calculated, and then the averages of the precision and recall values for each category are taken. The test results of the YOLOv5-CFD model are shown in Table 7.

The experiments of foreign fibers (including mulch film, foam, feather, white paper, polyethylene, polypropylene, and chemical fiber) detection and classification were made. The results showed that the environmental light intensity changes had some influence on the foreign fiber classification, but little effect on the detection. The interference of strong light such as sunlight caused an increase in the misrecognition rate. The classification performance of the model was the best under dark conditions and the worst under sunlight conditions. Foreign fibers were difficult to identify with a small or large incidence angle such as 15° or 90°. When the incident angle was around 45°, the detection and classification of foreign fibers were optimal. For the different variety of samples, the YOLOv5-CFD model could generally detect foreign fibers well, and the average F1 score of the three numbered samples was about 0.69. Under the condition of different positions of foreign fibers, there were no omissions and misjudgments, and the classification results were the same. Under the condition of different sizes of foreign fibers, the minimum size of foreign fibers detected by the YOLOv5-CFD model was 0.5 mm^2^. Smoke and dust almost had no interference of linear laser polarization imaging. In summary, the proposed method has good robustness and anti-interference, meets the basic detection of cotton foreign fibers on the actual production line, and has practical application value.

## 4. Conclusions

To address the problem of foreign fiber detection in cotton, a polarization imaging device of cotton foreign fiber was constructed using the difference in optical properties and polarization characteristics between cotton fibers and foreign fibers. Moreover, an object detection algorithm for cotton foreign fiber based on the improved YOLOv5 was proposed, which consisted of three key steps: The lightweight network Shufflenetv2 with the Hard-Swish activation function was used as the backbone feature extraction network, an improved PANet was added to YOLOV5, and a CA module was added before the Head of YOLOv5. The robustness and anti-interference of the improved YOLOv5 model under various conditions were also tested. Compared with the YOLOv5 foreign fiber detection model, the improved YOLOv5 foreign fiber detection model had better performance in mAP@0.5, mAP@0.5:0.95, and FPS, which increased by 1.03%, 7.13%, and 126.47%, respectively. The improved model is capable of performing online identification and classification of small foreign fiber targets of various types in cotton transportation. 

## Figures and Tables

**Figure 1 sensors-23-04415-f001:**
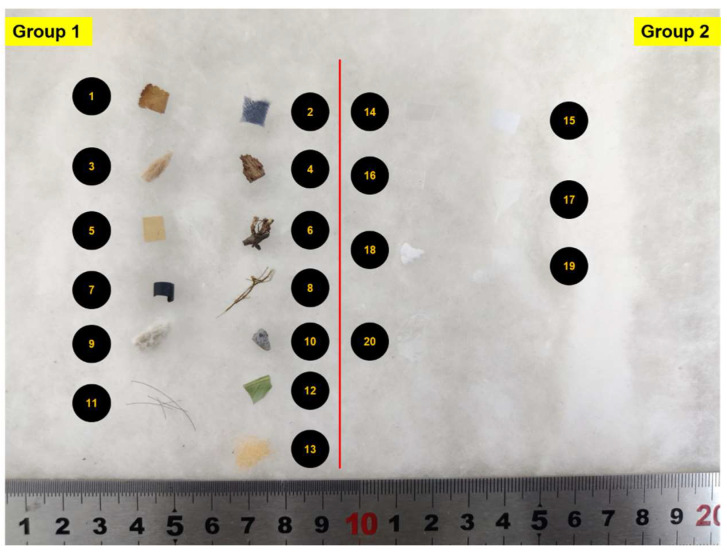
Foreign fiber and cotton samples. (1) Dead leaves; (2) cloth; (3) hemp rope; (4) bark; (5) kraft paper; (6) stalk; (7) PVC; (8) yarn; (9) cottonseed; (10) stone; (11) hair; (12) leaf; (13) sponge; (14) polypropylene; (15) white paper; (16) polyethylene; (17) feather; (18) foam; (19) chemical fiber; (20) mulch film.

**Figure 2 sensors-23-04415-f002:**
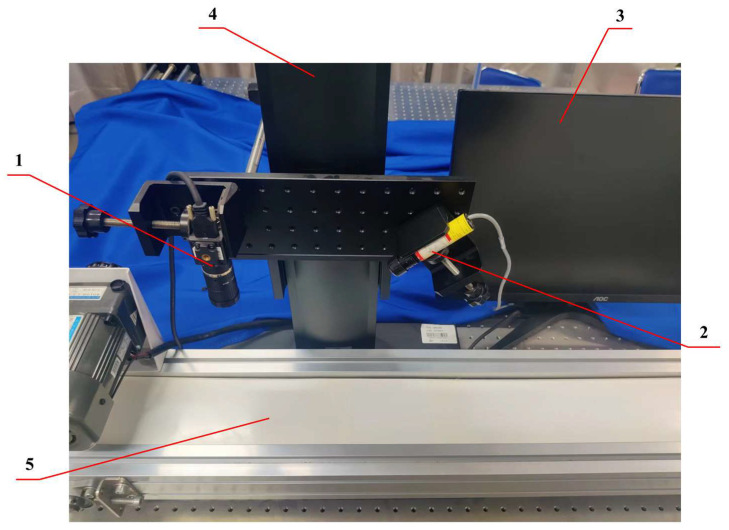
Physical image of cotton foreign fiber polarization imaging detection device. (1) Polarized camera; (2) 405 nm line laser; (3) computer; (4) machine vision frame; (5) electric conveyor belt.

**Figure 3 sensors-23-04415-f003:**
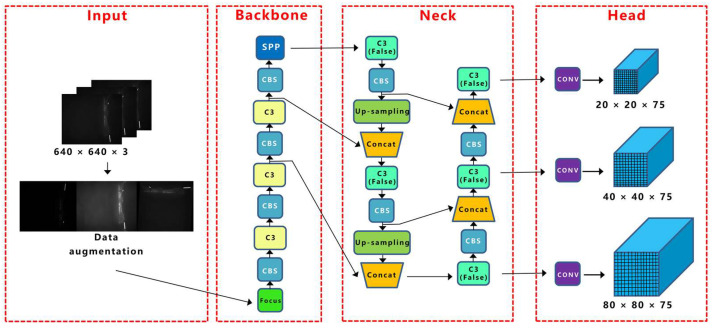
YOLOv5 network structure.

**Figure 4 sensors-23-04415-f004:**
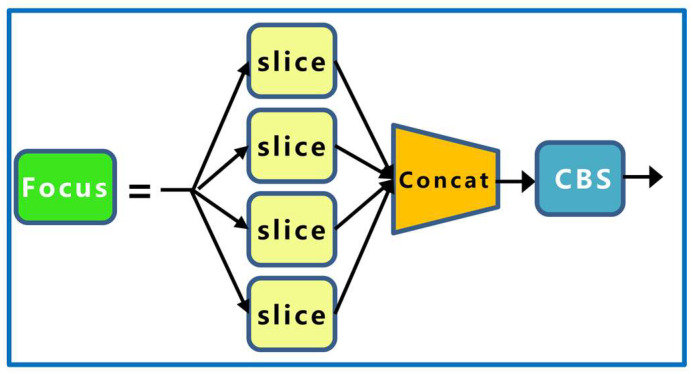
Focus network structure.

**Figure 5 sensors-23-04415-f005:**
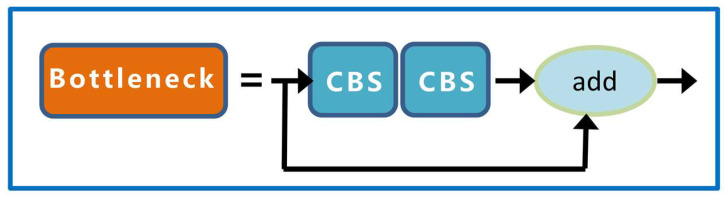
Bottleneck network structure.

**Figure 6 sensors-23-04415-f006:**
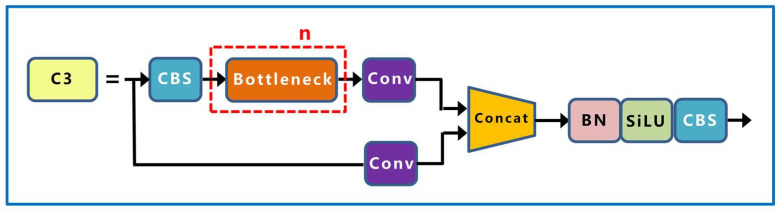
C3 network structure.

**Figure 7 sensors-23-04415-f007:**
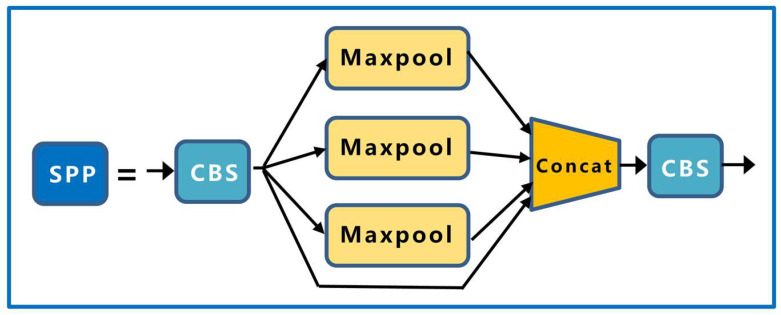
SPP network structure.

**Figure 8 sensors-23-04415-f008:**
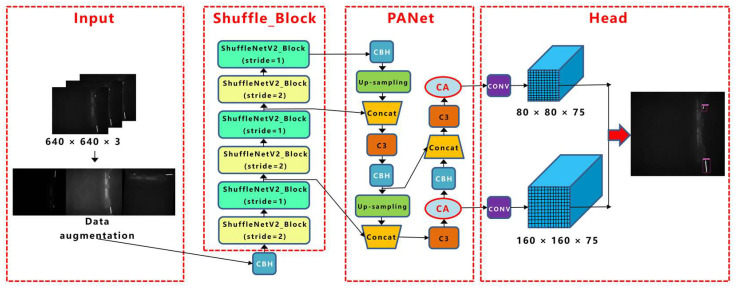
Network model of cotton foreign fiber detection.

**Figure 9 sensors-23-04415-f009:**
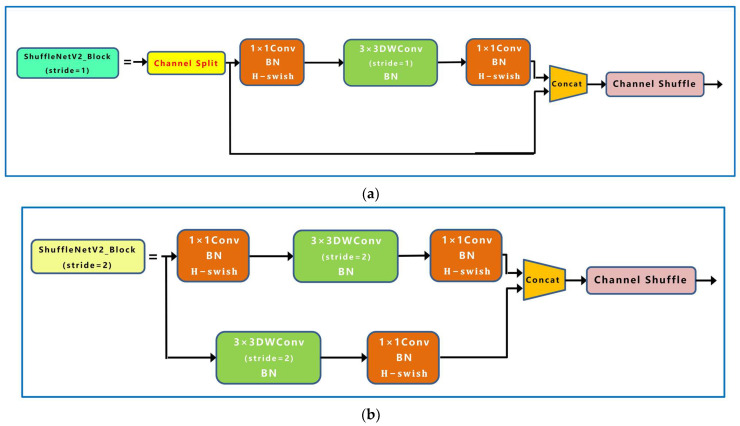
ShufflenetV2 network structure. (**a**) Basic unit; (**b**) unit for spatial downsampling (2×).

**Figure 10 sensors-23-04415-f010:**
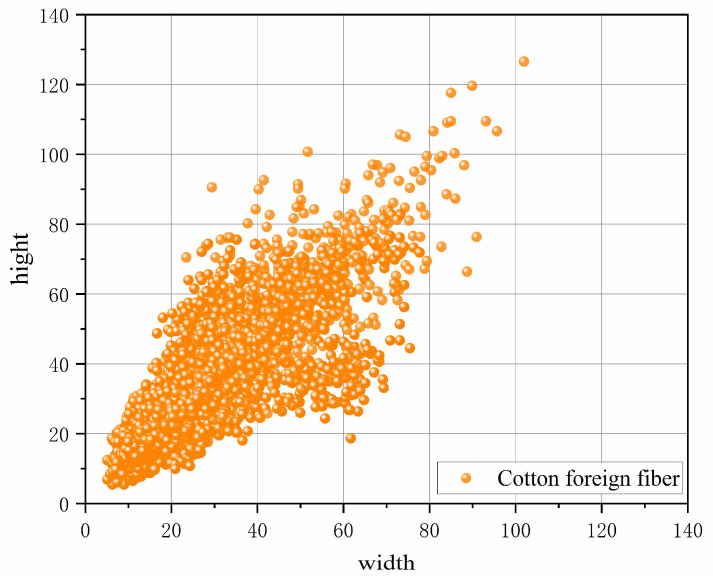
Cotton foreign fiber width and height distribution.

**Figure 11 sensors-23-04415-f011:**
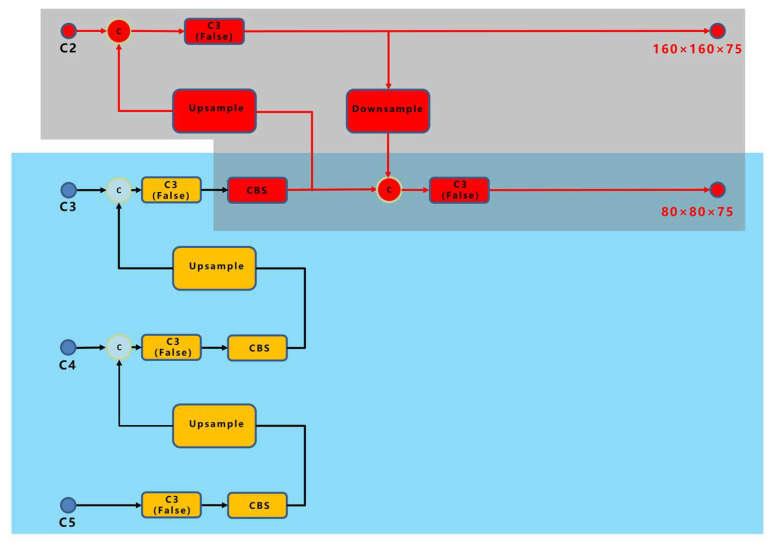
PANet network improvement schematic diagram of YOLOv5.

**Figure 12 sensors-23-04415-f012:**
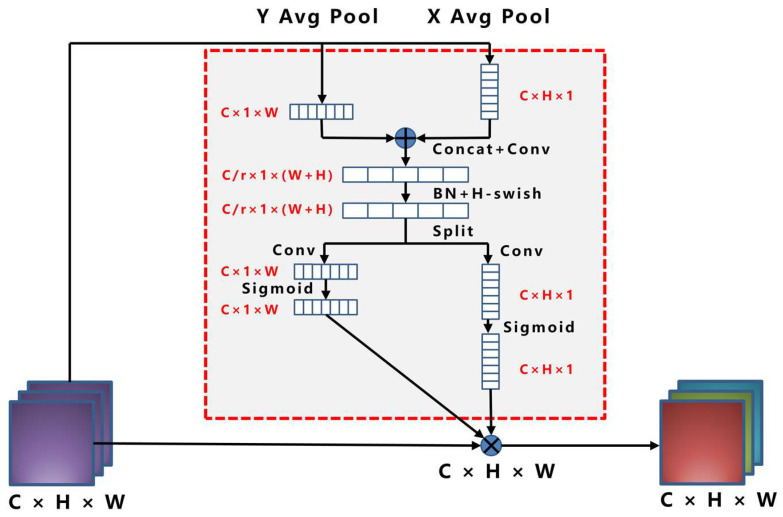
Coordinate Attention network structure.

**Figure 13 sensors-23-04415-f013:**
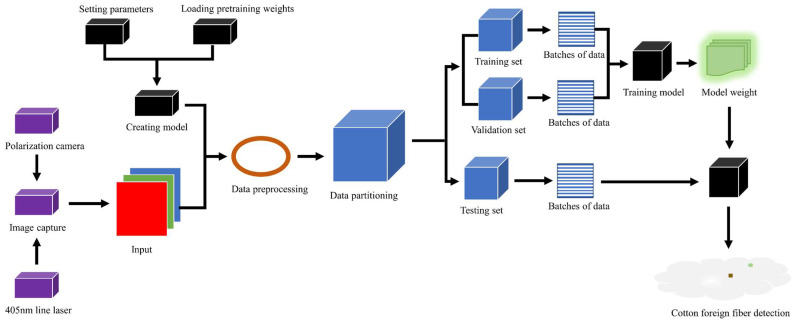
Flow chart of the proposed approach of foreign fiber detection.

**Figure 14 sensors-23-04415-f014:**
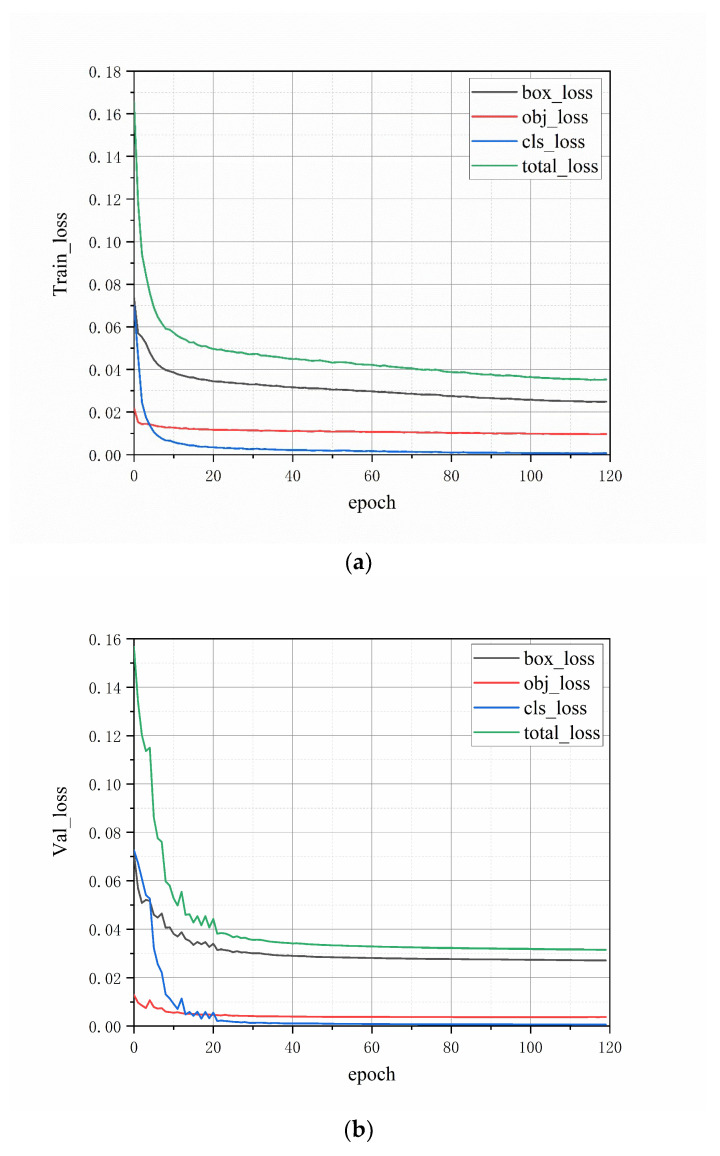
Four types of loss curves for each data set. (**a**) Training set; (**b**) Validation set.

**Figure 15 sensors-23-04415-f015:**
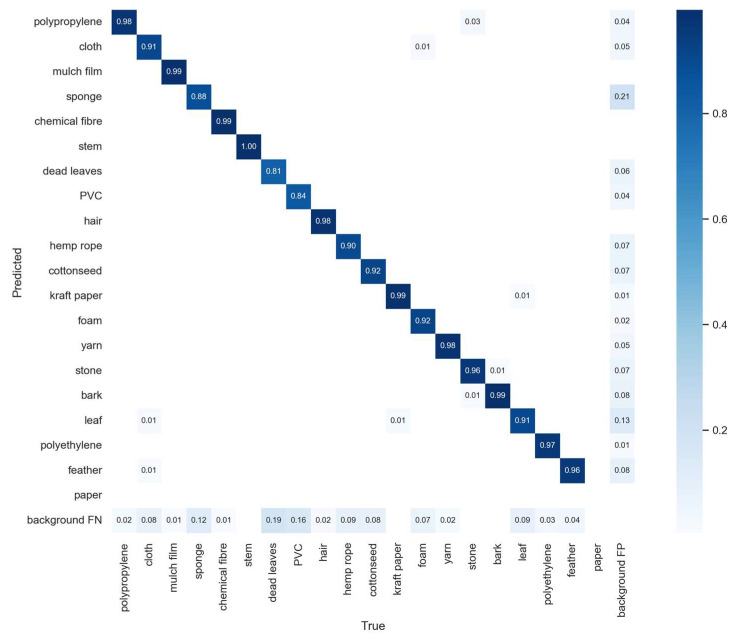
Confusion matrix of the YOLOv5-CFD model.

**Figure 16 sensors-23-04415-f016:**
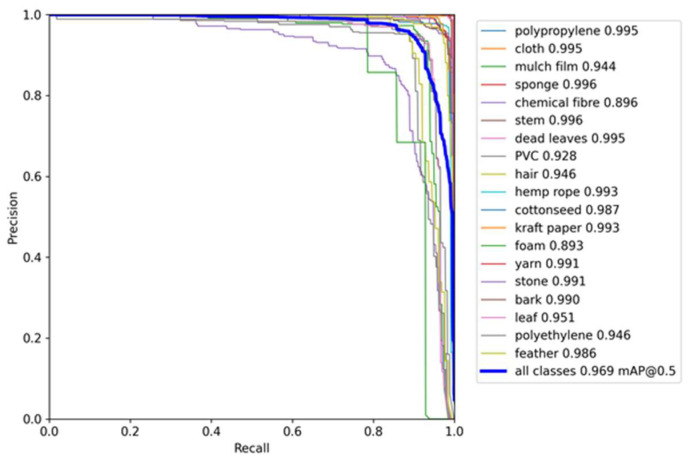
Test accuracy and recall curves of YOLOv5-CFD.

**Figure 17 sensors-23-04415-f017:**
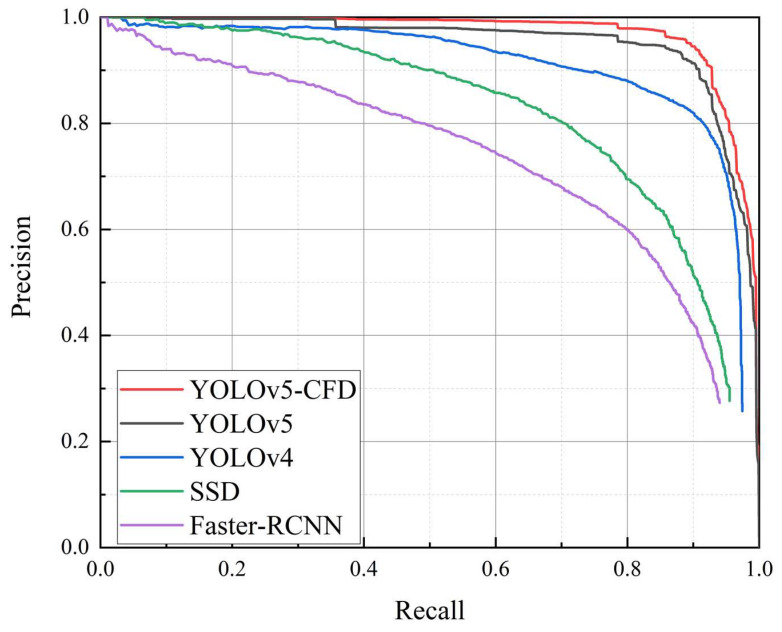
P–R curves of different detection models.

**Figure 18 sensors-23-04415-f018:**
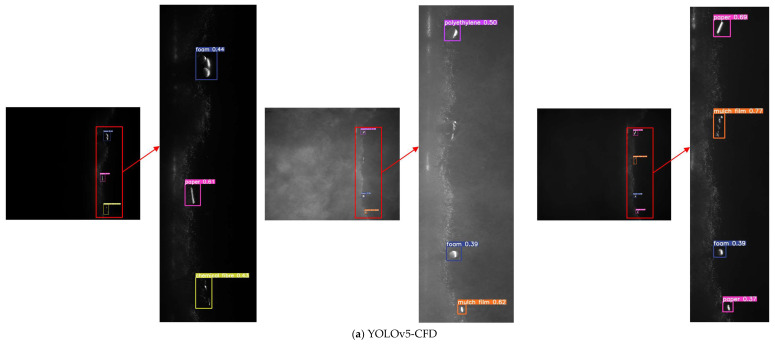
Visualization of different models in performance testing.

**Table 1 sensors-23-04415-t001:** Average gray value and contrast value of foreign fiber.

Parameter	Angle of Incidence			
	45°	60°	75°	90°
M(X)	85.0056	87.1309	79.0224	96.0166
Var(x, y)	5866.0319	5287.1670	4263.1522	2042.0076

**Table 2 sensors-23-04415-t002:** Target statistics of the cotton foreign fiber.

Categories	Number of Targets		
	Training Set	Validation Set	Testing Set
Polypropylene	967	112	121
Cloth	966	132	126
Mulch film	551	56	59
Sponge	954	109	137
Chemical fiber	976	111	113
Stem	994	108	98
Dead leaves	1002	102	114
PVC	1005	88	113
Hair	499	59	72
Hemp rope	1025	91	120
Cotton seed	989	108	121
Kraft paper	429	54	51
Foam	972	113	139
Yarn	108	11	16
Stone	974	109	117
Bark	991	101	108
Leaf	968	105	127
Polyethylene	985	115	136
Feather	987	120	117
Paper	975	121	134
Total	17,317	1925	2139

**Table 3 sensors-23-04415-t003:** Experimental environment configuration.

Hardware and Software	Configuration Parameter
Computer	Operating System: Windows10
	CPU: Intel(R) Core (TM) i9-9900K CPU@3.60GHz
	GPU: NVIDIA GeForce RTX 3090
	RAM: 16 GB
	Video memory: 24 GB
Software version	Python3.9.12 + PyTorch1.9.1 + CUDA11.7 + cuDNN8.2.1 + Opencv4.5.5 +Visual Studio Code2022 (1.69.1)

**Table 4 sensors-23-04415-t004:** Training network parameters.

Parameter	Value
Batch size	64
Learning rate	0.01
Warm-up epochs	3
Number of iterations	120
Momentum parameter	0.937
Image size	640 × 640
Optimizer	SGD

**Table 5 sensors-23-04415-t005:** Ablation experiments.

ShuffleNet V2	PANet	CA	H-Swish	Model Volume (MB)	mAP@0.5 (%)	mAP@0.5:0.95 (%)	FPS (f/s)
×	×	×	×	13.82	95.87	52.77	170
√	×	×	×	7.93	93.92	50.04	370
×	√	×	×	10.52	96.14	57.40	323
×	×	√	×	13.90	95.98	55.08	180
√	√	√	√	0.75	96.90	59.90	385

**Table 6 sensors-23-04415-t006:** Experimental results of different algorithms.

Model	Parameters	Model Volume (MB)	mAP@0.5 (%)	mAP@0.5:0.95 (%)	FPS (f/s)
YOLOv5-CFD	2.97 × 10^5^	0.75	96.90	59.90	385
YOLOv5	7.28 × 10^6^	13.82	95.87	52.77	170
YOLOv4	6.39 × 10^7^	244.78	93.59	50.50	88
SSD	2.41 × 10^7^	100.29	83.07	39.06	128
Faster-RCNN	2.84 × 10^7^	108.91	75.68	33.60	9

**Table 7 sensors-23-04415-t007:** YOLOv5-CFD model test results.

Different Conditions		Identification		Classification		
		Misrecognition Rate	Misjudgment Rate	Precision	Recall	F1
Illumination	dark	0	0	90.07%	97.03%	0.93
	lamplight	0	0	84.42%	92.35%	0.88
	sunlight	7.30%	0	72.20%	87.72%	0.79
Incidence angle	15°	100%	13.64%	0	0	0
	45°	0	0	90.07%	97.03%	0.93
	90°	3.93%	22.07%	67.85%	76.20%	0.72
Different varieties samples	115,549	0	5.54%	73.66%	66.23%	0.69
	114,835	0	7.92%	75.39%	70.03%	0.73
	114,712	12.87%	0	68.55%	60.34%	0.64
Different positions	upper edge	0	0	90.07%	97.03%	0.93
	middle	0	0	90.07%	97.03%	0.93
	lower edge	0	0	90.07%	97.03%	0.93
Foreign fiber size	1.5 mm^2^	0	0	70.79%	77.05%	0.74
	1 mm^2^	0	0	66.54%	78.21%	0.72
	0.5 mm^2^	16.05%	0	60.10%	71.78%	0.65
	<0.5 mm^2^	100%	0	0	0	0
Environment	smog	0	0	90.07%	97.03%	0.93
	dust	0	0	90.07%	97.03%	0.93

## Data Availability

Not applicable.

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
