# Peer review of "Detection and Classification of Cotton Foreign Fibers Based on Polarization Imaging and Improved YOLOv5"

_sensors, 2023, doi:10.3390/s23094415_

Round 1
Reviewer 1 Report
The authors proposed a deep-learning framework using an improved YOLOv5 for the detection and classification of cotton foreign fibers. It is interesting and I have the following comments.
1. What are the disadvantages of existing methods apart from requiring high-performance GPU?
2. What is the effect on the system performance for motion artifacts from the camera or conveyor belt?
3. The mathematical expression for ReLU should be provided.
4. Correction required in Figure 12, for 'hight'.
5. How gaussian blur is used for enhancement and what are the enhancement transformations used?
6. Eq. (9) should need correction.
7. What was the loss function used? and the mathematical expression should be included in the manuscript.
8. How did you compute the recall and precision for the multi-class? Complete details should be provided.
9. The objects in Figure 18 are too small to understand, some portions can be zoomed in.
10. What was the use of the validation set since you seem to handpick the hyperparameters of the model?
11. Multi-class confusion matrix should be provided on the results along with specificity and Matthew's correlation coefficient.
12. The code for the proposed work should be provided for review.
13. How much economic burden it can reduce, can the framework be extended to other types of materials?
Author Response
Dear Editor and Reviewers
First of all, we would like to thank you for your insightful comments and constructive suggestions, which will be helpful for improving our paper. According to your suggestions, the manuscript has been revised carefully. All the changes and additions have been marked by red color in the revised manuscript. The improvements are listed as bellow.
- Replies to the questions proposed by Reviewer 1
Reviewer #1---Q#1: What are the disadvantages of existing methods apart from requiring high-performance GPU?
Reply: Thank you for your comment. Although the proposed method in this paper can effectively detect surface foreign fibers of cotton layers, its detection performance is not satisfactory for foreign fibers inside cottons. Because experimental setup’s detection speed is limited by the camera sampling frequency, the model proposed in this paper can process 385 images per second. In future studies, we will adopt high-speed line-scan cameras to maximize the detection speed advantage of the model.
Reviewer #1---Q#2: What is the effect on the system performance for motion artifacts from the camera or conveyor belt?
Reply: Thank you for your comment. The motion artifacts from the camera or conveyor belt can cause image blur, thus affecting the accuracy of foreign fiber detection. Therefore, the detection system in this study fixed both the camera and conveyor belt, and the thickness of the cotton layer being inspected was around 2 mm, so that minor vibrations would not affect the clarity of the acquired images.
Reviewer #1---Q#3: The mathematical expression for ReLU should be provided.
Reply: Thank you for your comment. According to your suggestions, the formula for the H-swish activation function was replaced with a specific mathematical expression, as follows:
Before modification:
After modification:
The revision is incorporated in Eq. (9) of the revised manuscript.
Reviewer #1---Q#4: Correction required in Figure 12, for 'hight'.
Reply: Thank you for your comment. According to your suggestions, the 'height' in Figure 12 has already been corrected.
The revision is incorporated in Fig. 10 of the revised manuscript.
Reviewer #1---Q#5: How gaussian blur is used for enhancement and what are the enhancement transformations used?
Reply: Thank you for your comment. The Gaussian noise is a random number whose probability density function obeys a Gaussian distribution. The noise value is closely related to σ. Higher σ makes the data more widely distributed. By adding the Gaussian noise, the raw data is slightly changed, which becomes the noise data. The data sets are doubled by simply adding Gaussian noise to the samples (Improving machine learning based phase and hardness prediction of high-entropy alloys by using Gaussian noise augmented data[J]. Computational Materials Science, 2023, 223: 112140; Cassava disease recognition from low‐quality images using enhanced data augmentation model and deep learning[J]. Expert Systems, 2021, 38(7): e12746).
In this study, the iaa.GaussianBlur() function provided by the imgaug library in Python was utilized for Gaussian blur augmentation. The specific formula used was iaa.GaussianBlur(sigma=(0, 3.0)), where the parameter sigma represents the strength of the Gaussian blur, with sigma=0 indicating no blur and sigma=3.0 indicating strong blur. The value of sigma in this study was randomly selected between 0 and 3. Additionally, affine transformation (iaa.Affine()), brightness transformation (iaa.AddToBrightness()), pixel dropout (iaa.Dropout()), and flip augmentation (iaa.Fliplr()) were also employed.
The revision is incorporated in the revised manuscript on line no 160-161.
Reviewer #1---Q#6: Eq. (9) should need correction.
Reply: Thank you for your comment, we feel sorry for the mistake. The format of Eq. (9) has been corrected.
The revision is incorporated in Eq. (6) of the revised manuscript.
Reviewer #1---Q#7: What was the loss function used? and the mathematical expression should be included in the manuscript.
Reply: Thank you for your comment. In this paper, the functions of confidence loss, bounding box regression loss, and classification loss (, ……) are adopted, and the specific mathematical expressions and textual descriptions are included in the revised manuscript.
The revision is incorporated in the revised manuscript on line no 174-190.
Reviewer #1---Q#8: How did you compute the recall and precision for the multi-class? Complete details should be provided.
Reply: Thank you for your comment. In this paper, there are 20 types of foreign fibers. When considering a single category, such as the mulch film, it is treated as the positive class, and all other categories are treated as negative classes. Therefore, based on the revised manuscript's Eq. (4) and (5), the precision and recall values for each category can be calculated. In the revised Table 7, for each test condition, first calculate the precision and recall values for each category, and then take the average of the precision and recall values for each category.
The revision is incorporated in the revised manuscript on line no 455-457.
Reviewer #1---Q#9: The objects in Figure 18 are too small to understand, some portions can be zoomed in.
Reply: Thank you for your comment. Figure 18 has been modified, and the region of the foreign fiber detection box has been enlarged.
The revision is incorporated in Fig. 18 of the revised manuscript.
Reviewer #1---Q#10: What was the use of the validation set since you seem to handpick the hyperparameters of the model?
Reply: Thank you for your comment. In this study, the SGD (stochastic gradient descent) method was used to optimize the learning rate of the hyperparameters, and the epochs in the hyperparameters was determined by comparing the loss functions of the training set and validation set. We have provided corresponding supplementary explanations in the revised manuscript.
The revision is incorporated in the revised manuscript on line no 169-171.
Reviewer #1---Q#11: Multi-class confusion matrix should be provided on the results along with specificity and Matthew's correlation coefficient.
Reply: Thank you for your comment. The figure of multi-class confusion matrix and corresponding text have been added to the revised manuscript.
The revision is incorporated in the revised manuscript on line no 351-354 and Fig. 15.
Reviewer #1---Q#12: The code for the proposed work should be provided for review.
Reply: Thank you for your comment. If it is convenient, please provide me with an email address and we will discuss and exchange.
Reviewer #1---Q#13: How much economic burden it can reduce, can the framework be extended to other types of materials?
Reply: Thank you for your comment. In China, the price of foreign fiber detection equipment, such as the "Surpass" series produced by Daheng Image Company, is around 800,000 RMB. Our foreign fiber detection system has a total cost of around 100,000 RMB, which can significantly reduce economic burden. Our framework can also be applied to defect detection in textiles and silk.
Thank you for your consideration.

Reviewer 2 Report
This manuscript presented a foreign fibers detection and classification method based on an improved YOLOv5 algorithm and polarization imaging technology.
There are some issues needed to be addressed:
1. In the introduction, the foreign fibers classification method, especially based on deep learning, was not given enough references.
2. The experimental details should be described, such as the cotton sample size, foreign fiber varieties used and sizes, etc.
3. Table 7 should be described clearly to detect and classify. The classification results suitable for which kind of foreign fibers should be explained in detail.
4. English language and style are minor spell check required. Please correct it.
Author Response
Dear Editor and Reviewers
First of all, we would like to thank you for your insightful comments and constructive suggestions, which will be helpful for improving our paper. According to your suggestions, the manuscript has been revised carefully. All the changes and additions have been marked by red color in the revised manuscript. The improvements are listed as bellow.
- Replies to the questions proposed by Reviewer 2
Reviewer #2---Q#1: In the introduction, the foreign fibers classification method, especially based on deep learning, was not given enough references.
Reply: Thank you for your comment. References of foreign fiber classification method based on deep learning have been supplemented in the introduction of the revised manuscript.
The revision is incorporated in the revised manuscript on line no 81-92.
Reviewer #2---Q#2: The experimental details should be described, such as the cotton sample size, foreign fiber varieties used and sizes, etc.
Reply: Thank you for your comment. The cotton layer width, thickness, and the size of the foreign fibers have been added to the revised manuscript.
The revision is incorporated in the revised manuscript on line no 124-125, 135-136.
Reviewer #2---Q#3: Table 7 should be described clearly to detect and classify. The classification results suitable for which kind of foreign fibers should be explained in detail.
Reply: Thank you for your comment. Table 7 has been described clearly to detect and classify, and which foreign fibers are suitable in the classification results have been supplemented.
The revision is incorporated in the revised manuscript on line no 467-468 and Tab.7.
Reviewer #2---Q#4: English language and style are minor spell check required. Please correct it.
Reply: Thank you for your comment, the English language has been corrected.
Thank you for your consideration.

Reviewer 3 Report
Reviewers Comments
Major revision is being suggested for the manuscript id: sensors-2342211, titled, “Detection and Classification of Cotton Foreign Fibers Based on Polarization Imaging and Improved YOLOv5”. The following are specific comments; the author must revise the manuscript and prepare a rebuttal to the comments for further review.
1. Abstract: Line no 22-27: the sentences from “Moreover” onwards need to be rewritten with the quantitative claims to strengthen the last sentence of the abstract.
2. The introduction is too long, author can trim it, so the reader should not lose the interest.
3. Section 3, line no 204: is it needed to have such a separate section, as per my understanding it is a part of the materials and methods, so it can be easily explained as a subsection under section 2: materials and methods.
4. 4. Experimental Results and Analysis: the section heading should be results and discussion.
5. The data set is the part of materials and methods not the results part. Kindly shift that sub section Table-2 and table -3 to materials and methods section.
6. Shift the section 4.2 to materials and methods section.
7. Check equation no (9) for the typo mistake.
8. Kindly add one flow chart at the end of the section 2: materials and method, it must present complete methodology deployed in this paper. It makes a clear understanding for the readers.
9. Results and discussion: Although the results are well presented, the interpretation of the developed model's behaviour lacks adequate rationale. It is recommended that the author read recently published articles in prestigious journals of similar nature to correctly identify the cause of this behaviour. Additionally, the necessary citations must be included to support their findings.
10. Conclusions must be rewritten in no more than 150 words. It must present the major findings with quantitative information.
I wish authors a great success.
Author Response
Dear Editor and Reviewers
First of all, we would like to thank you for your insightful comments and constructive suggestions, which will be helpful for improving our paper. According to your suggestions, the manuscript has been revised carefully. All the changes and additions have been marked by red color in the revised manuscript. The improvements are listed as bellow.
- Replies to the questions proposed by Reviewer 3
Reviewer #3---Q#1: Abstract: Line no 22-27: the sentences from “Moreover” onwards need to be rewritten with the quantitative claims to strengthen the last sentence of the abstract.
Reply: Thank you for your comment. The abstract has been rewritten with the quantitative claims (The model volume, mAP@0.5, mAP@0.5:0.95, and FPS of the improved YOLOv5 was up to 0.75MB, 96.9%, 59.9% and 385f/s, compared with YOLOv5, the improved YOLOv5 increased by 1.03%, 7.13% and 126.47% respectively,).
The revision is incorporated in the revised manuscript on line no 23-25.
Reviewer #3---Q#2: The introduction is too long, author can trim it, so the reader should not lose the interest.
Reply: Thank you for your comment. The introduction has been reduced from 124 lines to 87 lines.
Reviewer #3---Q#3: Section 3, line no 204: is it needed to have such a separate section, as per my understanding it is a part of the materials and methods, so it can be easily explained as a subsection under section 2: materials and methods.
Reply: Thank you for your comment. Section 3 has been revised as a subsection under Section 2: materials and methods.
The revision is incorporated in the revised manuscript on line no 198-199, 252, 263, 283, 308.
Reviewer #3---Q#4: 4. Experimental Results and Analysis: the section heading should be results and discussion.
Reply: Thank you for your comment. The chapter title 4. Experimental Results and Analysis has been changed to 3. Results and Discussion.
The revision is incorporated in the revised manuscript on line no 341.
Reviewer #3---Q#5: The data set is the part of materials and methods not the results part. Kindly shift that sub section Table-2 and table -3 to materials and methods section.
Reply: Thank you for your comment. Tables 2 and 3 have been transferred to the Materials and Methods section.
The revision is incorporated in the revised manuscript on line no 163 and 167.
Reviewer #3---Q#6: Shift the section 4.2 to materials and methods section.
Reply: Thank you for your comment. Section 4.2 has been moved to the section on materials and methods.
The revision is incorporated in the revised manuscript on line no 173.
Reviewer #3---Q#7: Check equation no (9) for the typo mistake.
Reply: Thank you for your comment, we feel sorry for the mistake. The format of Eq. (9) has been corrected.
The revision is incorporated in Eq. (6) of the revised manuscript.
Reviewer #3---Q#8: Kindly add one flow chart at the end of the section 2: materials and method, it must present complete methodology deployed in this paper. It makes a clear understanding for the readers.
Reply: Thank you for your comment. A flowchart has been added at the end of section 2: Materials and Methods.
The revision is incorporated in Fig.13 of the revised manuscript.
Reviewer #3---Q#9: Results and discussion: Although the results are well presented, the interpretation of the developed model's behaviour lacks adequate rationale. It is recommended that the author read recently published articles in prestigious journals of similar nature to correctly identify the cause of this behaviour. Additionally, the necessary citations must be included to support their findings.
Reply: Thank you for your comment. We read how prestigious journal articles explain the behavior of the developed model, so as to modify part of the text of our paper and quote related articles (Face Mask Wearing Detection Algorithm Based on Improved YOLO-v4. Sensors 2021, 21, 3263; Cattle body detection based on YOLOv5-ASFF for precision livestock farming. Computers and Electronics in Agriculture 2023, 204, 107579).
The revision is incorporated in the revised manuscript on line no 395-401, 433-438.
Reviewer #3---Q#10: Conclusions must be rewritten in no more than 150 words. It must present the major findings with quantitative information.
Reply: Thank you for your comment. The conclusion has been rewritten with quantitative information about 150 words.
The revision is incorporated in the revised manuscript on line no 485-497.
Thank you for your consideration.

Reviewer 4 Report
The authors present a vision-based quality inspection system to detect and classify cotton foreign fibres in this work. They propose the use of a YOLOv5 network and polarization imaging as a solution.
Although the detection/absence of foreign fibres is presented, I think that this problem should be addressed from the beginning as it is an important part of any quality inspection system. Otherwise, the reader may think that the system has been developed just to distinguish among different foreign fibres.
I think that the introduction needs a deep review. The first paragraph extends from line 31 to line 89. In my opinion, the overall section lacks a proper structure, and some statements need to be rewritten to make them clearer. In addition, some references are too general. As an example, I can’t see the point of the sentence starting in line 112.
Extra editing must be carried out. For example, Table 1 is split into two different pages, and figures 2 and 3 are repetitive (consider merging them into a single one), Figure 3 caption is in a different page, figure 4 is difficult to read, …
Review equation 9.
Author Response
Dear Editor and Reviewers
First of all, we would like to thank you for your insightful comments and constructive suggestions, which will be helpful for improving our paper. According to your suggestions, the manuscript has been revised carefully. All the changes and additions have been marked by red color in the revised manuscript. The improvements are listed as bellow.
- Replies to the questions proposed by Reviewer 4
Reviewer #4---Q#1: Although the detection/absence of foreign fibers is presented, I think that this problem should be addressed from the beginning as it is an important part of any quality inspection system. Otherwise, the reader may think that the system has been developed just to distinguish among different foreign fibres.
Reply: Thank you for your comment. In the processes of cotton cultivation, harvesting, transportation, and processing, a large amount of foreign fibers are inevitably mixed in due to various factors, among which white and transparent fibers that are similar in color to cotton are difficult to detect. Therefore, it is particularly important to detect and remove foreign fibers in a timely manner in the initial stage before spinning, in order to ensure the quality of textile products. This article mainly focuses on detecting foreign fibers before spinning.
The revision is incorporated in the revised manuscript on line no 31-33.
Reviewer #4---Q#2: I think that the introduction needs a deep review. The first paragraph extends from line 31 to line 89. In my opinion, the overall section lacks a proper structure, and some statements need to be rewritten to make them clearer. In addition, some references are too general. As an example, I can’t see the point of the sentence starting in line 112.
Reply: Thank you for your comment. The introduction has been thoroughly reviewed and revised.
The revision is incorporated in the revised manuscript on line no 65-73.
Reviewer #4---Q#3: Extra editing must be carried out. For example, Table 1 is split into two different pages, and figures 2 and 3 are repetitive (consider merging them into a single one), Figure 3 caption is in a different page, figure 4 is difficult to read, …
Reply: Thank you for your comment, we feel sorry for the mistake. In the revised manuscript, Table 1 has been put on one page, Figure 2 and Figure 3 have been merged into one page, Figure 3 and the title have been put on one page, Figure 4 has been clearly shown, and related similar issues have been thoroughly reviewed.
Reviewer #4---Q#4: Review equation 9.
Reply: Thank you for your comment, we feel sorry for the mistake. The format of Eq. (9) has been corrected.
The revision is incorporated in Eq. (6) of the revised manuscript.
Thank you for your consideration.

Round 2
Reviewer 1 Report
No comments
Reviewer 3 Report
Reviewer’s Comments
The manuscript titled, “Detection and Classification of Cotton Foreign Fibers Based on Polarization Imaging and Improved YOLOv5”, with the Manuscript id: sensors-2342211, is updated as per the provided comments. I recommend that the manuscript be accepted.
I wish the authors great success.
